# A High Performance Triboelectric Nanogenerator Based on MXene/Graphene Oxide Electrode for Glucose Detection

**DOI:** 10.3390/ma16020841

**Published:** 2023-01-15

**Authors:** Wei Yang, Xu Cai, Shujun Guo, Long Wen, Zhaoyang Sun, Ruzhi Shang, Xin Shi, Jun Wang, Huamin Chen, Zhou Li

**Affiliations:** 1College of Materials and Chemical Engineering, Minjiang University, Fuzhou 350108, China; 2Fujian Key Laboratory of Functional Marine Sensing Materials, Minjiang University, Fuzhou 350108, China; 3Beijing Institute of Nanoenergy and Nanosystems, Chinese Academy of Sciences, Beijing 100083, China

**Keywords:** triboelectric nanogenerator, MXene, graphene oxide, nanostructure, glucose detection

## Abstract

A smart sensing platform based on a triboelectric nanogenerator (TENG) possesses various advantages such as self-powering, convenience, real-time and biocompatibility. However, the detection limit of the TENG-based sensor is required to be improved. In this study, a high performance TENG-based glucose sensor was proposed by using the Ti_3_C_2_T_x_ (MXene)/graphene oxide (GO) composite electrode. The MXene and GO nanosheets are popular 2D materials which possessed high conductivity and a rich surface functional group. The MXene/GO thin films were prepared through electrostatic self-assembly technology, which can effectively impede the agglomeration of two nanoflakes. The as-prepared MXene/GO film presented outstanding mechanical property. To figure out the relationship between the nanostructure of MXene/GO film and the TENG, a series of MXene/GO-based TENG with different GO sizes was characterized. As a result, the TENG with 400 nm GO demonstrated the highest output performance. Subsequently, the optimized TENG was used in glucose detection application without the assistance of a glucose enzyme. This simple and flexible TENG shows promising potential in biosensors and non-invasive health monitoring.

## 1. Introduction

The triboelectric nanogenerator (TENG) has become a promising candidate for the next generation of energy harvesting [1,2] and self-powered sensing platforms [3,4,5,6]. It possesses outstanding characteristics such as high energy conversion efficiency [7,8], abundant materials resource [9,10], a simple fabrication process [11,12] and multifunctional integration [13,14]. Since being invented in 2012 [15], it has made substantial improvements by various approaches including materials design [16,17], architecture modification [18,19] and concept development [20,21]. Especially, TENG-based sensors have attracted the wide attention of researchers [22]. The sensing mechanism is based on the output change of TENG caused by the detected object. The detected objects, such as heavy metal ions [23], dopamine [24], ethanol [25] and others, can react with the triboelectric materials [26,27,28], hence changing the output performance. The advantages of TENG-based ion detection are that it is self-powered, convenient, real-time and biocompatible. However, the detection limit of the TENG-based sensor is required to be improved.

The group of 2D materials composed of transition metal carbides and nitrides is called MXene (Ti_3_C_2_T_x_), which possesses high conductivity and a controllable surface functional group. It has been widely used in TENG and various sensors [29,30,31,32]. On the one hand, MXene has strong electronegativity and excellent conductivity [33], which makes it a promising triboelectric material. The MXene-based TENG shows high performance and flexibility for wearable applications [34]. On the other hand, the rich surface terminations such as -F, -OH and -O greatly influence the electrical, mechanical and electrochemical properties [35]. In particular, MXene has become an emerging material for smart sensing on account of its electrochemical property [36]. It has been applied to gas sensors [37], electrochemical sensors [38] and biosensors [39]. Pure MXene has limited mechanical property owing to the self-stacking effect. To decrease self-stacking, other nanomaterials such as polymers [40], nanowires [41], nanofibers [42] and graphene oxide (GO) [43] are introduced to form MXene-composite films. Among these, MXene/GO composite materials have exhibited extraordinary performance in supercapacitors [44], batteries [45] and flexible sensors [46]. Nevertheless, the MXene/GO-based TENG for ion detection was rarely reported.

In this work, we proposed a high performance TENG based on a MXene/GO composite film for glucose detection. The MXene/GO films were prepared by vacuum-assisted filtration of mixed MXene/GO suspension. By electrostatic self-assembly technology, the MXene/GO film showed excellent electrical and mechanical properties. This is because the interaction between the GO and MXene effectively impedes the agglomeration of two nanoflakes and improves the conductivity and electrochemical activity. Afterwards, we adjusted the size of the GO to form MXene/GO films with different nanostructures. Additionally, the relationship between the output performance of the MXene/GO-based TENG and the nanostructure of the composite film was investigated. As a result, the MXene/GO-based TENG with 400 nm GO flakes exhibited optimal performance. Its open-circuit voltage, current output and surface charges were 258.8 V, 2 μA and 37.8 nC. Finally, the MXene/GO-based TENG was demonstrated to detect glucose concentration. The abundant surface radicals bind to glucose and change the triboelectric ability. This will result in an obvious output performance change at a low range of glucose. The flexible MXene/GO-based TENG shows outstanding potential in non-invasive sensing and self-powered physical health monitoring.

## 2. Materials and Methods

### 2.1. Preparation of MXene Dispersion and GO Aqueous Solution

The Titanium aluminum carbide (Ti_3_AlC_2_) powder (particle size: 0.2–10 μm) was purchased from Jilin 11 Technology Co., Ltd. A total of 1.0 g lithium fluoride (LiF) was dissolved into 16 mL of hydrochloric acid (HCl) solution (9 mol/L) as an etching precursor. Then, 0.45 g Ti_3_AlC_2_ powder was added. After 48 h of reaction time at 40 °C, the acquired product was centrifuged with HCl solution (6 mol/L) and deionized water three times, respectively, until the PH value was close to 6. Then, the mixture was treated by ultrasonic wave for 40 min and centrifuged for 5 min (4000 rpm) to remove the residual precipitation, and the colloidal solution of a few-layered MXene dispersion was obtained.

The monolayer GO aqueous solutions (1 mg/mL) was purchased from Nanjing/China XFNANO Materials Co., Ltd. The sizes of GO flakes were 50–200 nm (small size), <500 nm (medium size) and >500 nm (large size), respectively.

### 2.2. Fabrication of MXene/GO Composite Electrode

The MXene dispersion and GO aqueous solution were mixed by ultrasonic treatment for 30 min to obtain MXene/GO dispersion. The volume ratios of MXene/GO with different sizes were all 1:1. Afterwards, the MXene/GO thin films were fabricated by electrostatic self-assembled technology through vacuum-assisted filtration. Then, the MXene/GO films were dried at 60 °C for 2 h in air. The as-prepared MXene/GO electrodes can be easily peeled off from the filter membrane (Polypropylene). These samples were labeled MG_S_, MG_M_, MG_L_, which corresponded to the various GO sizes. In addition, the thickness of the MXene/GO films was controlled by the volume (20 mL) of the dispersion, and the thickness was around 10 μm in the next experiment.

### 2.3. Construction of Flexible MXene/GO-Based TENG

Firstly, the Ecoflex (volume ratio of A: B, 20mL, Smooth-on, Inc., Macungie, PA, USA) was spin-coated on a glass substrate at a speed of 300 rpm for 20 s and cured at 120 °C for 3 h. A copper (Cu) conductive tape was pasted on the back of the Ecoflex with a thickness of 1 mm. The Cu electrode and Ecoflex film acted as the bottom electrode and triboelectric layer. The MXene/GO film was adhered to a acrylic plate (3M Company, Saint Paul, MN, USA) with a diameter of 6 cm. The two films were connected to the two separated plates of a test instrument.

### 2.4. Characterization

The surface morphologies was characterized by a field-emission scanning electron microscope (SEM, SU8000, Hitachi, Tokyo, Japan). The X-ray diffraction (XRD) pattern was acquired by the diffractometer (MiniFlex600, Rigaku, Tokyo, Japan). The size of the nanosheet was evaluated by a nano particle size analyzer (NanoPlus3). A linear motor control system (LSP1, China) was utilized to provide the contact forces and frequencies. The resistance was characterized by four-point probes (FT-331, China), and the electric performance output of TENG was recorded by an electrometer (Keithley 6517B).

### 2.5. Glucose Detection

To evaluate the sensitivity of MXene/GO-based TENG for glucose detection, the glucose was dissolved into water to obtain a glucose solution with a concentration of 1 mol/L. Constant volumes of 0.1 mL were dropped onto the surface of MG_M_ electrode, followed by drying in a natural environment. Then, the MGM-based TENG was tested by a linear motor with a contact force of 50 N under a frequency of 2 Hz.

## 3. Results and Discussion

The MXene/GO electrodes were fabricated following the above process displayed in Figure 1a. The Ti_3_AlC_2_ powder was etched by LiF and HCl solution to obtain Ti_3_C_2_T_x_ MXene colloidal solution. Then, the acquired product was followed by centrifugation, ultrasonic vibration, centrifugation and drying treatment, in that order. Subsequently, the MXene nanosheets were used to form MXene dispersion with a concentration of 1 mg/mL. Afterwards, the GO dispersion with different GO sizes was mixed with the MXene dispersion by ultrasonic treatment to obtain stable MXene/GO dispersion. The volume ratio of MXene/GO was 1:1. Thereafter, the MXene/GO thin films were fabricated by electrostatic self-assembled technology through vacuum-assisted filtration. The as-prepared freestanding MXene/GO electrodes were labeled MG_S_, MG_M_, MG_L_, which corresponded to the various GO sizes. The diameters of the MXene/GO films were all 5.5 cm. Finally, the MXene/GO electrodes were utilized to construct the MXene/GO-based TENG. The structure illustration was exhibited in Figure 1b. The MXene/GO electrode acted as the top electrode and triboelectric layer, simultaneously. Considering the electronegativity of MXene, Ecoflex was selected as the bottom triboelectric layer, and Cu conductive tape was attached to the Ecoflex, which acted as the bottom electrode. In addition, as the interaction between the GO and MXene effectively impedes the agglomeration of two nanoflakes, the mechanical strength of MXene/GO film is greatly improved. The flexibility of MXene/GO electrode was shown in Figure 1c.

The morphology and electrical property of the MXene/GO electrodes were characterized in Figure 2. The flexible and freestanding MG_S_, MG_M_ and MG_L_ films all showed metallic luster, which was fabricated by the above-mentioned vacuum-assisted filtration. As shown in Figure 2a–c, the surfaces of the three MXene/GO electrodes were very smooth, and the MG_M_ film was darker. The diameter of the MXene/GO electrodes were 6 cm. There were three different GO nanosheet samples, which were 50–200 nm, <500 nm and >500 nm, respectively. The high-resolution SEM images of various GO nanosheets were compared in Figure 2d–f. Seen from these images, the GO nanosheets were about 200 nm, 400 nm and 1 μm, respectively. Additionally, a dynamic light scattering (DLS) method was used to evaluated the size of the GO nanosheet, as shown in Appendix A. According to the DLS measurement, the hydrodynamic diameters of different GO nanosheets were 233 nm, 2038 nm and 2533 nm, respectively. The high-resolution image of the few-layered MXene was presented in Figure 2g. The maximum size of the MXene nanosheet was about 5 μm, and most of the MXene nanosheets were smaller than 1 μm. The sizes of the MXene nanosheet and the GO nanosheet are comparable. Additionally, the intercalation of MXene nanosheets can effectively impede the self-agglomeration. The cross-sectional SEM and surface SEM images of the MXene/GO electrode were shown in Figure 2h,i. The thickness of the electrode was around 10 μm. The phase transition of the materials was characterized by XRD. As shown in Figure 2j, the characteristic of (002) peak at 2*θ* = 6.7° for the MXene films clearly demonstrated the few-layered MXene. The peak at 2*θ* = 9.5° for the GO film was corresponding to the (001) plane, and the peak at 2*θ* = 22.6° for the GO film can be attributed to the reduced GO nanosheets introduced in the heat process. However, the intensity of these peaks of the MXene/GO films apparently dropped, which indicates the MXene and GO nanosheets are effectively embedded into each other. In addition, the sheet resistance and the conductivity of the MG_S_, MG_M_ and MG_L_ films are revealed in Figure 2k,l. The MG_M_ film demonstrated the highest conductivity.

The working mechanism of the MXene/GO-based TENG was illustrated in Figure 3a. In the original state, the MXene/GO film and the Ecoflex film were separated, and there were no triboelectric charges. Under external pressure, the two films were in close contact, and there were equal triboelectric charges (i). After releasing the external pressure, the potential difference between the two electrodes will result in current flow from the Cu electrode to the MXene/GO electrode until potential equilibrium (ii). As the distance keeps increasing, the electrons continuously flow from the top electrode to the bottom electrode (iii). As the external pressure is applied again, the direction of the electrons’ flow is reversed until the two films contact again (iv). The periodic contact and separation process can result in an AC current. The potential distribution of the MXene/GO-based TENG at the open-circuit state is simulated in Figure 3b. Firstly, the distance between the two films is nearly zero, and there is nearly no potential difference between the MXene/GO electrode and the Cu electrode. As the distance is increased to 5 mm, the potential difference increased to around 100 V. Then, the potential difference reaches around 200 V as the two films returns to the original state. The potential difference is proportional to the distance between the two films. It is worth noting that the internal resistance of Keithley 6517B is 200 TΩ. Thus, the measured output voltage is approximate to the open-circuit voltage.

The triboelectric performance of the MXene/GO-based TENGs was evaluated by using a TENG test system consisting of a linear motor, Keithley 6517B and a data acquisition board. The external force, working frequency and distance between the two films were controlled by the system. Firstly, the relationship between the size of the GO nanosheets and the electrical performance of the TENG was investigated. Three TENGs based on MG_S_, MG_M_ and MG_L_ were fabricated, and the open-circuit voltage of corresponding samples was compared in Figure 4a. These TENGs were measured under a force of 50 N with a frequency of 2 Hz. The MG_M_-based TENG possessed the highest open-circuit voltage of 258.8 V, which was larger than the voltage of the MG_S_-based TENG (148.9 V) and the MG_L_-based TENG (187.2 V). This can be attributed to the electrostatic self-assemble with suitable GO size. Smaller or larger size will influence the self-assembled process. This is consistent with the above materials characterization. In addition, the current output in Figure 4b and surface charges in Figure 4c presented a similar trend. The current output of TENGs based on MG_S_, MG_M_ and MG_L_ were 1.5 μA, 2.0 μA and 1.3 μA, respectively, whereas the corresponding surface charges were 22.0 nC, 37.8 nC, and 28.7 nC, respectively. As the MG_M_-based TENG presented the highest electrical output, it was chosen for the next investigation.

Moreover, the effect of external force on the output performance of the TENG was studied at a frequency of 2 Hz, and the results were compared in Figure 4d–f. Evidently, the output performance of TENG increased with increasing external force. This phenomenon is on account of closer contact at the larger force, resulting in more triboelectric charges. For example, the surface charges increased from 28.9 nC to 40.6 nC as the external force increased from 10 N to 60 N. Additionally, the open-circuit voltage and current output can reach 268.7 V and 2.3 μA under a force of 60 N. Furthermore, the relationship between the frequency and the output performance was exhibited in Figure 4g–i. These TENGs were measured under a force of 60 N. Clearly, the surface charges and open-circuit voltage was insensitive to the frequency. This phenomenon has been explained in our previous work [47,48]. The surface charges and voltage output were measured in the open-circuit state, and the frequency has no effect on the triboelectric charges. On the contrary, the current output increased rapidly from 1.6 μA to 5.5 μA with the increasing frequency of 1 Hz to 5 Hz. The frequency affects the flow process of electrons, thus resulting in the current enhancement. To sum up, the output performance can be further improved by controlling the parameters such as external force and frequency.

Power supply capability is also an important index for the TENG in practical applications. The load resistance is various in the actual circuit board; therefore, it is necessary to measure the output power at different resistances. The result is presented in Figure 5a. As can be seen, the output power increased from nearly 0 to 84.5 μW with increasing load resistance from 10 kΩ to 100 ΩM. Then, it was dramatically decreased as the load resistance was further increased. Subsequently, the MG_M_-based TENG was used to charge various capacitors. As presented in Figure 5b, the 1 μF capacitor can be charged to 16 V in 200 s, and even the 10 μF capacitor can be charged to around 3 V in 200 s. This output is sufficient to power microelectronic devices. Furthermore, the potential for the long-term work of the TENG was assessed in Figure 5c. After continuously working for 2400 cycles (40N, 2Hz), the open-circuit voltage was stable around 250 V. Despite there being about a 15% drop in the open-circuit voltage, the voltage was still 234.2 V. The stability test further confirmed the mechanical strength of MXene/GO films.

To estimate the biosensor capability of the as-prepared TENG, the common and important substances including glucose and Na^+^ were investigated. Especially, the glucose concentration in blood and human extracorporeal fluid is a significant clinical index, which is closely related to diabetes and hypoglycemia. In the detection experiment, constant volumes of 0.1 mL (1 mol/L) were dripped onto the surface of the MG_M_ electrode, followed by drying in a natural environment for 5 min. A shielded cover was used to avoid contamination. Then, the electrical performance of the MG_M_-based TENG was conducted under a 50 N force and a 2 Hz frequency. The open-circuit voltages of the MG_M_-based TENG at various glucose concentrations were displayed in Figure 6a. Apparently, the voltage is monotonically correlated to the glucose concentration in the range of 0–500 μmol. The output voltage increased from 258.4 V to 426.8 V when increasing the glucose concentration to 500 μmol. The MXene-based TENG via the filtration method [33,49] was used as a control experiment to figure out the effect of the GO nanosheet. As seen from Appendix A, the initial output voltage of the MXene-based TENG was much lower than that of the MG_M_-based TENG. Additionally, the output voltage tends to be saturated at a higher glucose concentration. The MG_M_-based TENG shows higher sensitivity and a wider detection range. Compared with the MXene film, the MXene/GO film has abundant channels and surface terminations. The absorption of glucose will adjust the electronegative property. In addition, the effect of the Na^+^ ion on the output voltage was explored in Figure 6b. The relationship between the output voltage and the Na^+^ concentration is ambiguous. The voltage fluctuation is probably because the water molecule enters the porous structure, which slightly influences the electron gain/loss capacity. The sensitivity of the as-prepared TENG for glucose and Na^+^ ion detection were shown in Figure 6c,d. Different from traditional enzyme-based glucose sensor, the MXene/GO-based TENG utilizes the reactive functional group on the surface. Furthermore, a comparison with other TENG-based sensors has been undertaken, which is shown in Appendix A.

## 4. Conclusions

In summary, we proposed a high performance MXene/GO-based TENG for non-invasive glucose detection. Through an electrostatic self-assembly process, the MXene/GO film effectively impeded the self-agglomeration, resulting in improved mechanical and electrochemical properties. Subsequently, we adjusted the nanostructure of the MXene/GO film by introducing various GO sizes, which were labelled as MG_S_, MG_M_ and MG_L_. Additionally, the relationship between the output performance of the MXene/GO-based TENG and the nanostructure of composite film was investigated. As a result, the MG_M_-based TENG presented optimal performance. Its open-circuit voltage, current output and surface charges were 258.8 V, 2 μA and 37.8 nC, respectively. Finally, the MXene/GO-based TENG was demonstrated to detect glucose concentration. The abundant surface functional groups bind to glucose and change the triboelectric ability. This results in an obvious output performance change at a low range of glucose. Consequently, the flexible MXene/GO electrode demonstrates promising potential in high performance TENG and paves the way for the development of biosensors and non-invasive health monitoring.

## Figures and Tables

**Figure 1 materials-16-00841-f001:**
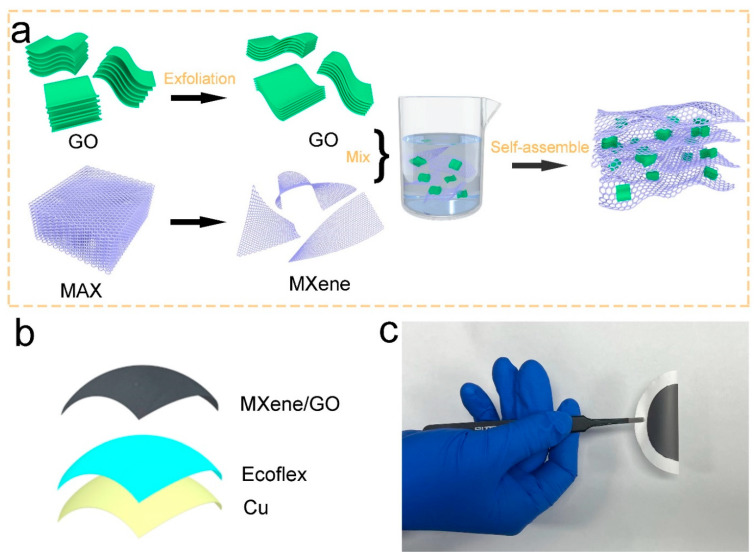
Schematic illustration of the MXene/GO thin film and the MXene/GO-based TENG. (**a**) The schematic fabrication process of the MXene/GO electrode. (**b**) Schematic structure of the MXene/GO-based TENG. (**c**) A photograph of the flexible MXene/GO film.

**Figure 2 materials-16-00841-f002:**
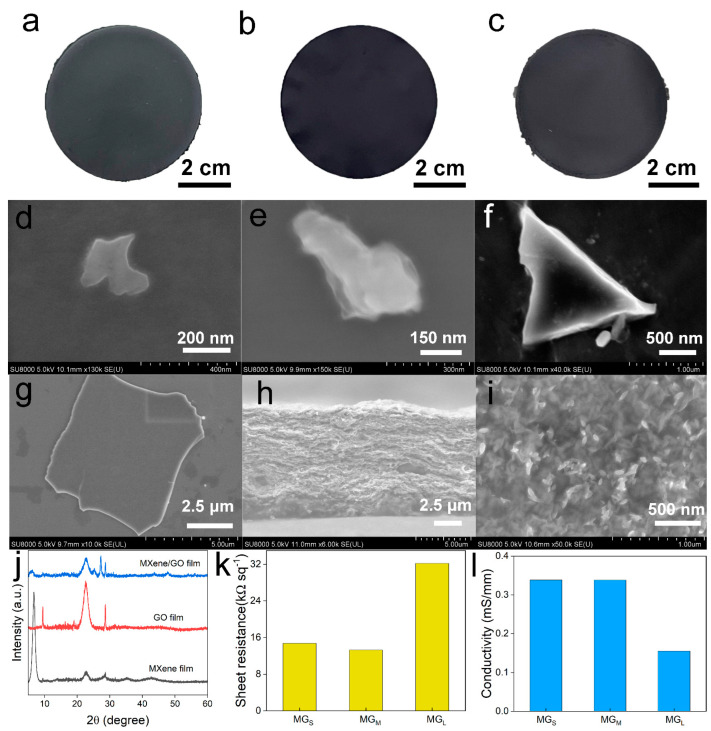
Characteristics of the MXene/GO films. (**a**–**c**) Photographs of MG_S_, MG_M_ and MG_L_. (**d**–**f**) SEM images of the GO nanosheets with different sizes. (**g**) SEM of the MXene nanosheet. (**h**) Cross-sectional SEM of the MXene/GO films. (**i**) SEM of the MXene/GO film. (**j**) The XRD pattern of the MXene, GO and MXene/GO films. (**k**) The sheet resistances of MG_S_, MG_M_ and MG_L_. (**l**) The conductivity of MG_S_, MG_M_ and MG_L_.

**Figure 3 materials-16-00841-f003:**
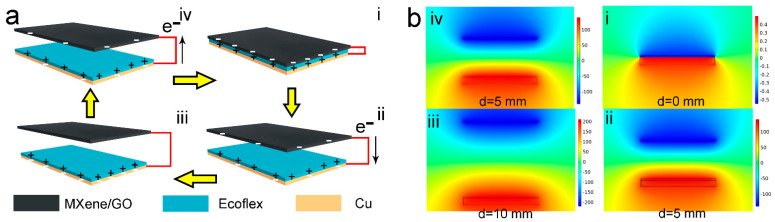
Working mechanism of the MXene/GO-based TENG. (**a**) The charge distribution and charge transfer in the contact-separation mode. (**b**) The potential difference between the MXene/GO film and the Cu film in the working process.

**Figure 4 materials-16-00841-f004:**
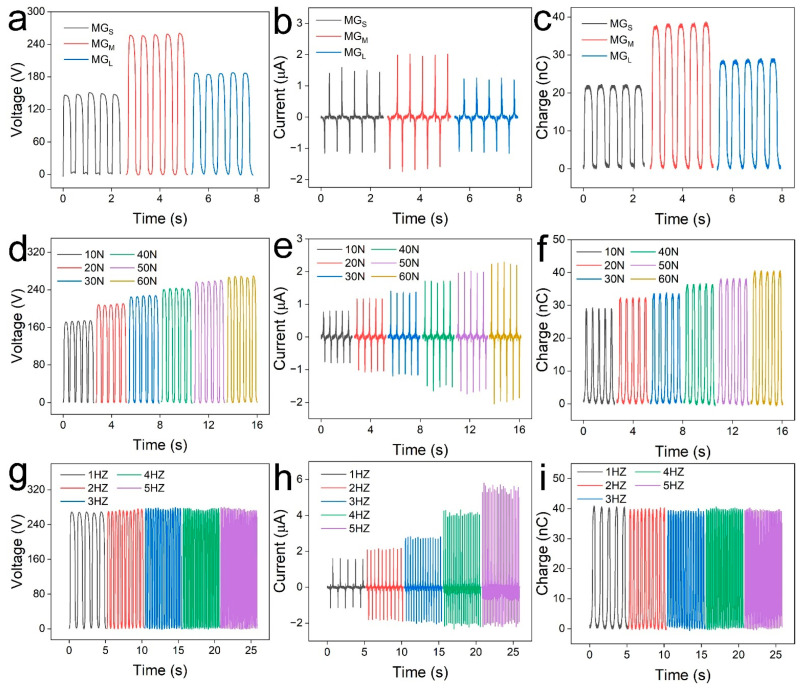
Electrical performance characterization of the MXene/GO-based TENG. (**a**) Open-circuit voltage, (**b**) current output and (**c**) surface charges of TENGs based on MG_S_, MG_M_ and MG_L_. The relationship of (**d**) open-circuit voltage, (**e**) current output and (**f**) surface charges of MG_M_-based TENG with the external force. The relationship of (**g**) open-circuit voltage, (**h**) current output and (**i**) surface charges with the working frequencies.

**Figure 5 materials-16-00841-f005:**
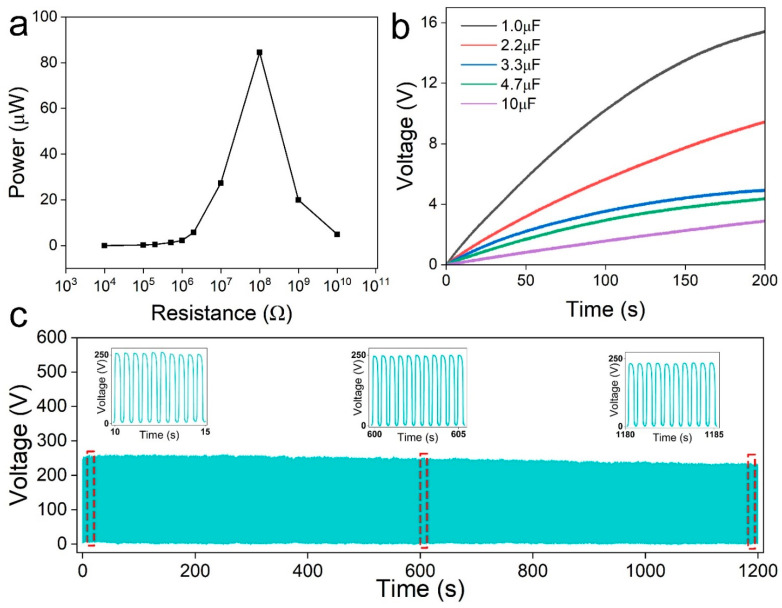
Powering capability of the MG_M_-based TENG. (**a**) The output power under various load resistance. (**b**) Charging curve of various capacitors (1 μF, 2.2 μF, 3.3 μF, 4.7 μF and 10 μF). (**c**) The stability test of about 2400 cycles.

**Figure 6 materials-16-00841-f006:**
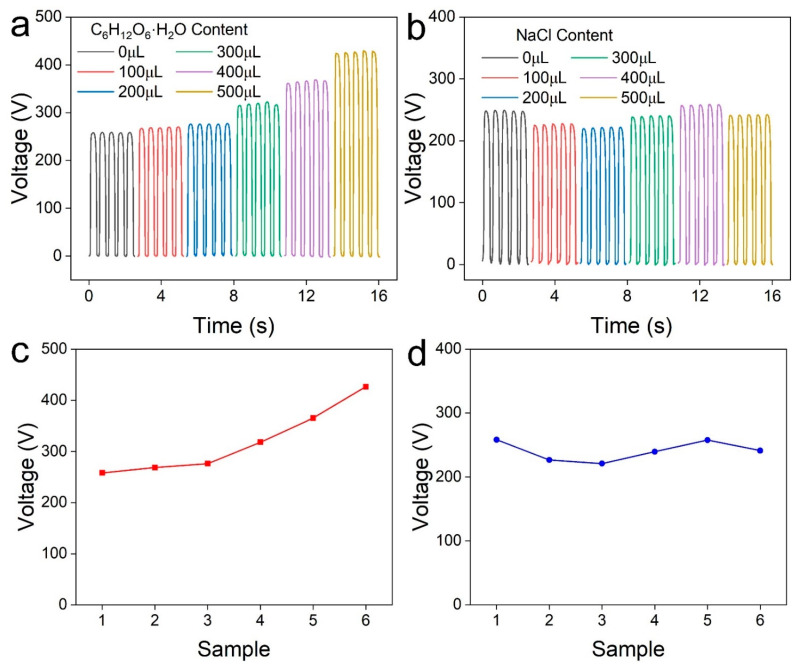
Characterization of ion detection of MG_M_-based TENG. The open-circuit voltage of MG_M_-based TENG at various (**a**) glucose concentrations and (**b**) salt solutions. The relationship between the voltage and (**c**) the glucose concentrations. (**d**) The salt solution.

## Data Availability

The data presented in this study are available on request from the corresponding author.

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
