# Peer review of "A High Performance Triboelectric Nanogenerator Based on MXene/Graphene Oxide Electrode for Glucose Detection"

_materials, 2023, doi:10.3390/ma16020841_

Round 1
Reviewer 1 Report
The authors report the development of a triboelectric nanogenerator based on MXene/GO thin films for glucose monitoring. The reported results could have relevance in biosensing field. However, I believe that before this paper can be considered for publication in this Journal, some revisions are need.
- The detection of glucose was performed only on MXene/GO-based TENG while the performance of the TENG sensor should be evaluated also in the absence of GO. Flexible MXene composed triboelectric nanogenerator via facile vacuum-assistant filtration method have been already synthesized (see. Z. - - Zhang et al., Nano Energy, 2021, 88, 106257). I believe that the advantages in adding GO should be discussed.
- The size of MXene and GO nanosheet was evaluated only by SEM images; I believe that in bulk studies such as DLS measurements are needed.
- The electrical performance of the TENG sensor should be compared with that obtained from other biosensors
Minor:
-The chemical composition of MXene must be defined were firstly written
-The scale bars of SEM images in figure 2 are not clearly visible and should be better highlighted.
Reviewer 2 Report
The article “A high performance triboelectric nanogenerator based on MXene/graphene oxide electrode for glucose detection” is a TENG-based glucose sensor that is proposed by using the MXene/graphene oxide (GO) composite electrode. The experiments are performed well but this work can not be accepted in its present form. Work can be considered for potential publications after the following major changes/clarifications:
1. The authors mentioned 2 nanoflakes clusters in the abstract if they can elaborate on it a bit later. It will help to attract more non-scientific audiences too.
2. Some sentences do not have clear meaning like “Subsequently, the optimized TENG was applied to glucose detection application without the help of glucose enzyme.”, it can be rephrased as Following that, the optimized TENG was used in glucose detection applications without the assistance of a glucose enzyme and so on.
3. Reduce the lengths of the sentences as the first sentence in the introduction is covering more than 4 lines, which is too much for remaining focused.
4. Kindly mention the chemicals once before using their formulas like HCL, Ti3AlC2, etc.
5. The line 50 - 200 nm (XF020), <500 nm (XF020), and> 500 nm (XF020), is quite confusing if it represents MGS, MGM, and MGL then mention next to it.
6. The authors mentioned that the samples were dried in the air, so what were precautions considered to avoid contamination?
7. Section 2.2 last line should be combined with the second last line also mention the volume to achieve 10-micron thickness.
8. The thickness of all MXene/GO films on acrylic was the same as 10 microns, if yes then please mention, also mention the source from which the acrylic was purchased.
9. Please mention the white color substrate shown in Figure 1(c).
10. The device was based on the single electrode, if yes then how the current and voltage was calculated?
11. Kindly add some details regarding the device's stability, reliability, and hysteresis effect.
12. I would like to advise that authors include some more references from the current year, as well as references from previous years like https://doi.org/10.1016/j.jallcom.2020.156702, etc.
Round 2
Reviewer 1 Report
I believe that this revised version of the manuscript can be accepted for publication in this Journal.
Reviewer 2 Report
Dear Authors,
The paper after changes seems acceptable from my side.
Best Regards